# A Systematic Review and Meta-Analysis of the Prevalence of Small Fibre Impairment in Patients with Fibromyalgia

**DOI:** 10.3390/diagnostics12051135

**Published:** 2022-05-03

**Authors:** Eleonora Galosi, Andrea Truini, Giulia Di Stefano

**Affiliations:** Department of Human Neuroscience, Sapienza University of Rome, Viale dell’ Università 30, 00185 Rome, Italy; eleonora.galosi@uniroma1.it (E.G.); giulia.distefano@uniroma1.it (G.D.S.)

**Keywords:** small fibre pathology, neuropathic pain, skin biopsy, fibromyalgia

## Abstract

Converging evidence shows that patients with fibromyalgia syndrome have signs of small fibre impairment, possibly leading to pain and autonomic symptoms, with a frequency that has not yet been systematically evaluated. To fill this gap, our review aims to define the frequency of somatic and autonomic small fibre damage in patients with fibromyalgia syndrome, as assessed by objective small fibre-related testing. We found 360 articles on somatic and autonomic small fibre assessment in patients with fibromyalgia. Out of the 88 articles assessed for eligibility, 20 were included in the meta-analysis, involving 903 patients with fibromyalgia. The estimated prevalence of somatic small fibre impairment, as assessed with skin biopsy, corneal confocal microscopy, and microneurography, was 49% (95% confidence interval (CI): 39–60%, I^2^ = 89%), whereas the estimated prevalence of autonomic small fibre impairment, as assessed with heart rate variability, sympathetic skin response, skin conductance, and tilt testing, was 45% (95% CI: 25–65%, I^2^ = 91%). Our study shows that a considerable proportion of patients with fibromyalgia have somatic and autonomic small fibre impairment, as assessed by extensive small fibre-related testing. Nevertheless, the heterogeneity and inconsistencies across studies challenge the exact role of small fibre impairment in fibromyalgia symptoms.

## 1. Introduction

Fibromyalgia is a chronic primary pain condition associated with autonomic symptoms, fatigue, and cognitive disturbances. It affects between 2% and 4% of the general population, though females are affected more often than males [1].

Despite the large body of studies on the topic, the mechanisms underlying this common chronic pain condition are still a matter of debate. For years, research pointed to central nervous system abnormalities [2,3,4]. However, recent preclinical investigations have suggested that dorsal root ganglia damage, due to circulating autoantibodies, may play a role in fibromyalgia [5,6]. Accordingly, different studies, using skin biopsy, corneal confocal microscopy, and microneurography, have shown that a variable proportion of patients with fibromyalgia have reduced intraepidermal nerve fibre density and abnormal C-fibre activity [6,7,8,9]. The small fibre impairment described in these studies, commonly defined as small fibre pathology [7], may, therefore, be associated with the pain and autonomic disturbances that patients with fibromyalgia commonly experience. The impact of small fibre impairment over distally distributed painful symptoms, which represent an independent factor affecting the life quality of patients with fibromyalgia [10], has yet to be defined. 

The frequency of small fibre impairment has previously been investigated in fibromyalgia syndrome, although the research is limited to skin biopsy and corneal confocal microscopy studies [7]. To the best of our knowledge, no systematic review has investigated the frequency of small fibre impairment as assessed by extensive small fibre-related objective testing for somatic and autonomic nerve fibres.

More precise information on the frequency of somatic and autonomic small fibre impairment in patients with fibromyalgia might further support the sparse evidence that small fibre pathology underlies pain and autonomic disturbances in patients with fibromyalgia.

The aim of this study was to provide information on the frequency of small fibre pathology in patients with fibromyalgia, as assessed by extensive small fibre-related objective testing. To do so, we systematically reviewed studies that used objective small fibre assessment and provided information on somatic and autonomic small fibre impairment in patients with fibromyalgia.

## 2. Methods

### 2.1. Search Process

We systematically reviewed and analysed literature adhering to the preferred reporting items for systematic reviews and meta-analyses (PRISMA) statements. PubMed, Embase and Cochrane Library databases were initially systematically searched on 3 December 2020. The primary search was supplemented by a secondary search using the bibliographies of the retrieved articles. The search strategy with specific combination search outcomes is outlined in the Appendix A. We limited the search to English language publications up to November 2021. Studies published as only an abstract were excluded. The target population included patients of any age with a diagnosis of fibromyalgia, according to widely agreed diagnostic criteria [11,12]. Studies including fewer than 10 patients with fibromyalgia were excluded. We included studies testing somatic small fibre impairment with objective methods, i.e., skin biopsy, corneal confocal microscopy, evoked potentials, and microneurography. Using the same approach, we systematically reviewed studies investigating autonomic small fibre impairment as assessed with heart rate variability (HRV), head-up tilt testing, quantitative sudomotor axon reflex testing (QSART), electrical skin conductance, sympathetic skin response, and laser Doppler imaging. For the meta-analysis, we included studies providing an estimated prevalence of small fibre impairment.

Risk of bias was assessed using the tool developed by Munn et al. [13]. A total score was calculated for each article, reflecting the overall risk of bias. A total score between 0 and 3 corresponded to a low risk, 4–6 to a moderate risk, and ≥7 indicated a high risk of bias. 

The review process was performed independently by two reviewers.

### 2.2. Meta-Analysis

Each selected article was included within a meta-analysis to determine the overall prevalence of small fibre impairment and autonomic disfunction in fibromyalgia; the 95% confidence interval (CI) associated with the prevalence value was calculated by using the Wilson score. 

We used the heterogeneity index I^2^ as a measure of the inconsistency between study results. In the case of high heterogeneity index values (>50% for overall pooled data and subgroup analyses), the pooled frequency and the relevant confidence interval (CI) were computed using a random effects model; otherwise, a fixed effects model was used. Per each pooled analysis (overall and subgroups), the total variance ${Χ}^{2}$, the variance between studies $\tau^{2}$, and the two-sided test for overall effect (Z, P) with a significance level of 95% were computed and reported. Calculations were carried out using Cochrane RevMan 5.4 [14].

## 3. Results

We identified 360 studies that assessed somatic and autonomic small fibre impairment in patients with fibromyalgia (Figure 1). After abstract screening, 88 full-text studies were assessed for eligibility. For the meta-analysis, we excluded 12 irrelevant studies, 53 studies lacking an estimated frequency of small fibre impairment, and two studies involving fewer than 10 patients. 

### 3.1. Somatic Small Fibre Impairment

We identified 31 studies investigating somatic small fibre impairment in patients with fibromyalgia. Of these 31 studies, 23 used skin biopsy, 4 used corneal confocal microscopy, 2 used microneurography, and 2 used laser-evoked potentials. 

All skin biopsy studies homogenously provided evidence of small fibre pathology in patients with fibromyalgia, and all reported that the intraepidermal fibre density was lower in patients with fibromyalgia than in healthy controls. Nevertheless, only 11 studies provided an estimated frequency of skin biopsy abnormalities (i.e., an intraepidermal nerve fibre density reduction), according to standard normative ranges [15,16,17,18,19,20,21,22,23,24,25]. 

When skin biopsies were taken from two sites of the leg, i.e., from above the lateral malleolus within the sural nerve territory and from the lateral upper thigh, a homogeneous reduction in intraepidermal nerve fibre density was found [16,25]. This finding is potentially consistent with small fibre impairment without a distal gradient, i.e., ganglionopathy. Most skin biopsy studies used protein gene product (PGP) 9.5 immunostaining to quantify all intraepidermal nerve fibres, though some studies also used alternative immunostaining. One study showed that the density of intraepidermal nerve fibres immunostained with growth-associated protein (GAP) 43 was lower in patients with fibromyalgia than in healthy controls, thus indicating that fibromyalgia is associated with lower regenerating nerve fibre density [25]. One study that collected skin biopsies from the glabrous hypothenar and trapezius regions found increased peptidergic sensory innervation of cutaneous arteriole-venule shunts, suggesting abnormal small blood vessel function through an altered neuropeptide response and the upregulation of adrenergic receptors [26]. A skin biopsy study found that the mean axon diameter was reduced in patients with fibromyalgia compared with patients with small fibre neuropathy and healthy controls [27]. These findings were in line with a previous electron microscopy study that investigated the skin of patients with fibromyalgia and found Schwann cell ballooning and peripheral axonal abnormalities, including smaller axons [28]. 

Skin biopsy studies that investigated skin cytokine expression in patients with fibromyalgia reported contradictory findings. While some studies found inflammatory changes and increased skin cytokine gene expression, thus suggesting that small fibre pathology is associated with inflammation [29,30], other studies found that skin cytokine gene expression did not differ between patients with fibromyalgia and controls [31].

Only two studies using corneal confocal microscopy reported an estimated frequency of abnormalities [8,32]. These two studies showed decreased nerve fibre length, density, and branching, compared to healthy controls. One study showed changes in corneal innervation and Langerhans cell distribution in patients with fibromyalgia and small fibre neuropathy, potentially based on altered Langerhans cell signalling [33]. Another study showed that basal epithelial cell density, total nerve density, long nerve fibres, and the number of nerves were lower in patients with fibromyalgia than in controls [34].

Two studies using microneurography reported abnormal somatic small fibre function, including spontaneous activity and sensitization to mechanical stimulation [9,18]. 

Eight studies tested laser-evoked potentials in patients with fibromyalgia. Two studies using laser-evoked potentials to detect small fibre pathology in fibromyalgia did not report small myelinated Aδ fibre function abnormalities [16,35]. Five studies showed increased amplitude of laser-evoked potentials and reduced habituation in patients with fibromyalgia, supporting an abnormal central elaboration of pain and pain matrix hyperexcitability [36,37,38,39]. In one study, the Aδ laser-evoked potential amplitude, conditioned by a preceding C-fibre laser-evoked potential, was significantly higher in patients with fibromyalgia than in healthy controls, supporting the hypothesis of pain matrix hyperexcitability [4].

For the meta-analysis, we included 14 studies (11 using skin biopsy, 2 using corneal confocal microscopy, 2 using laser-evoked potentials and 2 using microneurography), involving 769 patients with fibromyalgia, which provided an estimated frequency of somatic small fibre impairment (Table 1).

The estimated frequency of somatic small fibre impairment ranged between 0 and 85% (Figure 2). A forest plot analysis showed that the estimated prevalence, based on a random effects model, was 43% (95% CI: 26–61%), with a high level of heterogeneity (c^2^ = 874; t^2^ = 0.133; df = 16; I^2^ = 98%. Test for overall effect: Z = 4.798; *p* < 0.0001).

#### 3.1.1. Subgroup Pooled Analysis—Skin Biopsy

In this subgroup analysis, we included 11 studies using skin biopsy that had enrolled 591 patients [15,16,17,18,19,20,21,22,23,24,25]. The estimated frequency ranged between 30% and 85% (Figure 3). A forest plot analysis showed that the estimated prevalence of somatic small fibre impairment in patients with fibromyalgia, based on a random effects model, was 50% (95% CI: 37–63%), with a high level of heterogeneity (c2 = 119; t2 = 0.043; df = 10; I2 = 92%. Test for overall effect: Z = 7.537; p < 0.0001).

#### 3.1.2. Subgroup Pooled Analysis—Corneal Confocal Microscopy

Two studies tested somatic small fibre impairment with corneal confocal microscopy [8,32]. These two studies included 56 patients with fibromyalgia and provided an estimated frequency of 51% and 71%, respectively (Figure 4). A forest plot analysis showed that the estimated prevalence of somatic small fibre impairment in patients with fibromyalgia, based on a random effects model, was 60% (95% CI: 41–78%), with a moderately high level of heterogeneity (c^2^ = 2; t^2^ = 0; df = 1; I^2^ = 50%. Test for overall effect: Z = 5.873; *p* < 0.0001).

#### 3.1.3. Subgroup Pooled Analysis—Microneurography

In this subgroup analysis, we included two studies that tested somatic small fibre impairment with microneurography [9,18]. These studies included 57 patients with fibromyalgia and provided an estimated frequency of 30% and 41%, respectively (Figure 5). A forest plot analysis showed that the overall frequency of somatic small fibre impairment in patients with fibromyalgia was 35% (95% CI: 22–47%), with low heterogeneity (c^2^ = 1; t^2^ = 0.000; df = 1; I^2^ = 0%. Test for overall effect: Z = 5.540; *p* < 0.00001). In this case, a fixed effects model was used.

### 3.2. Autonomic Small Fibre Impairment in Fibromyalgia

We identified 45 studies that used heterogeneous methods to investigate autonomic small fibres in patients with fibromyalgia. Of these 45 studies, 29 used HRV analysis [40,41,42,43,44,45,46,47,48,49,50,51,52,53,54,55,56,57,58,59,60,61,62,63,64,65,66,67,68,69,70,71]. Of the 29 studies testing autonomic dysfunction with HRV analysis, three studies did not find significant differences between patients with fibromyalgia and controls [42,52,54,55], while 26 studies found abnormal HRV parameters in patients with fibromyalgia. Of these 26 studies, 8 demonstrated only frequency-domain impaired variables and found a homogeneously increased low-to-high frequency (LF/HF) ratio [40,45,46,50,59,61,66,70], while nine studies demonstrated only time-domain impaired variables [41,43,44,47,48,55,56,60,69] and nine studies demonstrated both frequency- and time-domain variable abnormalities [49,51,53,57,58,64,67,68,71]. Only two studies [42,49] that respectively provided a 40% and 83% estimated frequency of autonomic dysfunction with HRV were included in the meta-analysis.

Seven studies investigated skin conductance [40,72,73,74,75,76,77] and seven investigated sympathetic skin response [38,41,78,79,80,81,82]. Four showed that patients with fibromyalgia had reduced baseline skin conductance, as compared with healthy controls [40,72,73,74], while two studies [75,76] found no significant differences, and one study found higher conductance levels in patients with fibromyalgia [77]. All seven studies performed the test from the hand, with only one study also analysing the foot [73]. One study reported the frequency of sudomotor dysfunction, as assessed by skin conductance (28%) [73].

Seven studies tested autonomic small fibres using sympathetic skin response. Of these seven studies, three did not find significant differences between patients and healthy controls [41,78,80], while three studies found significantly prolonged sympathetic skin response latencies in patients with fibromyalgia [38,79,81], and two studies found a significantly reduced sympathetic skin response amplitude in patients with fibromyalgia, as compared with healthy controls [79,82]. Two studies [38,82] respectively provided an 18% and 39% estimated frequency of autonomic dysfunction with sympathetic skin response.

Three studies used head-up tilt testing and two demonstrated a higher frequency of tilt-test positivity in patients, with respect to healthy controls [62,67]. These two studies provided an estimated frequency of autonomic small fibre impairment of 43.7% and 64.7%, respectively.

Three studies tested laser Doppler flowmetry in patients with fibromyalgia. Two studies showed that patients with fibromyalgia took longer times to peak than healthy controls, indicating reduced blood flux, as assessed by measurement over the tender points [83,84]. Another study showed that patients with fibromyalgia had weaker responses to local heat (a vasodilator stimulus) than healthy controls [85]. 

No study used QSART, which is considered the gold standard for sudomotor function assessment. No study performed quantification of autonomic innervation of cutaneous annexes at skin biopsy, though one study analysed the innervation of arteriole-venule shunts [26].

#### Pooled Analysis of Autonomic Small Fibre Impairment Frequency

Of the 45 studies investigating small fibre impairment in patients with fibromyalgia, we included seven studies in the pooled analysis. These studies used heterogeneous methods to report the frequency of autonomic small fibre impairment in 141 patients with fibromyalgia [38,42,49,64,67,73,82]. Tilt testing was used in two studies [62,67], HRV analysis in two [42,49], skin conductance in one [73], and sympathetic skin response in two [38,82] (Table 2).

The estimated frequency of autonomic small fibre impairment ranged between 18% and 83% (Figure 6). A forest plot analysis showed that the estimated prevalence of autonomic small fibre impairment in patients with fibromyalgia, based on a random effects model, was 43% (95% CI: 26–60%), with a high level of heterogeneity (c^2^ = 35; t^2^ = 0.036; df = 6; I^2^ = 86%. Test for overall effect: Z = 5.003; *p* < 0.0001).

Risk of bias

The majority of studies had a low risk of bias. The risk of bias was calculated as moderate in two studies that did not provide a sample size calculation, involved a small sample of patients, or did not provide a detailed description of the statistical analysis and cut-off values [8,23].

## 4. Discussion

Our study shows that approximately 50% of patients with fibromyalgia have signs of somatic and autonomic small fibre impairment, lending further support to the evidence that small fibre pathology may contribute to the pain and autonomic symptoms that patients with fibromyalgia experience. 

Our systematic review defines the frequency of somatic and autonomic small fibre damage in patients with fibromyalgia syndrome, as assessed by extensive objective small fibre-related testing, including morphometric and functional tests for both somatic and autonomic fibre evaluation. To the best of our knowledge, previous studies [7] have assessed the frequency of somatic small fibre impairment as detected by morphometric measures, such as skin biopsy and corneal confocal microscopy, but no systematic review has provided evidence on small fibre pathology frequency as assessed by more extensive objective testing, including functional tests and autonomic nerve fibre assessment.

### 4.1. Somatic Small Fibre Impairment

All 31 skin biopsy studies included in our systematic review homogenously provided evidence of small fibre impairment in patients with fibromyalgia and consistently reported a lower intraepidermal fibre density in patients with fibromyalgia than in healthy controls. These findings support the current notion that fibromyalgia is associated with peripheral nervous system damage [7]. The 11 skin biopsy studies, involving 591 patients, included in this meta-analysis show that 50% of patients with fibromyalgia have a reduced intraepidermal nerve fibre density, compatible with small fibre pathology. This estimated frequency is in line with a recent meta-analysis [7] that included six studies and reported a 45% frequency of reduced intraepidermal nerve fibre density. 

The tests for statistical heterogeneity indicated significant variability in the overall skin biopsy data (I^2^ = 92%). This high heterogeneity likely reflects the different criteria for skin biopsy abnormalities across the included studies. While the diagnosis of small fibre pathology relied on reduced intraepidermal nerve fibre density at the distal site in some studies [16,23,24], other studies used different criteria and found higher frequencies of small fibre pathology. For instance, one study, which collected skin biopsies in three locations and diagnosed small fibre pathology when the intraepidermal nerve fibre density was abnormal in one of these three locations, found 76% prevalence of small fibre pathology [21]. 

We identified two studies that used corneal confocal microscopy to investigate small fibre pathology in patients with fibromyalgia, and provided information on the frequency of abnormalities [8,32]. The quality and reliability of the meta-analysis findings, however, are hampered by the relatively small sample of subjects included in the two studies (56 patients) and the different methodological approaches. While the study by Oudejans [32] relied on widely agreed parameters (i.e., nerve fibre length, nerve fibre density, and branching), and clearly reported diagnostic cut-off values, the study by Ramirez [8] investigated different parameters (stromal nerve thickness and corneal sub-basal plexus nerve density) and did not provide diagnostic cut-off values. 

Two studies used microneurography to investigate small fibre impairment in patients with fibromyalgia [9,18]. These two studies provided a relatively homogeneous estimated frequency of microneurographic abnormalities (30% and 41%), with low heterogeneity (I^2^ = 0%). However, the microneurographic recordings in these two studies were carried out by the same authors, and, thus, these findings should be replicated by independent groups. 

Unexpectedly, the small fibre impairments identified by most studies using skin biopsy, corneal confocal microscopy, and microneurography were not often supported by studies using functional assessments of the somatosensory system, such as quantitative sensory testing or laser-evoked potentials. Most studies using quantitative sensory testing [16,27] showed that patients with fibromyalgia had warm and cold detection thresholds (mediated by small fibres) that fell within normative ranges, while recent studies found no laser-evoked potential abnormalities in patients with fibromyalgia [35]. The inconsistency between the evidence of small fibre pathology and the lack of functional somatosensory abnormalities may raise the possibility that small fibre pathology contributes to the pain experienced by patients with fibromyalgia, in association with additional mechanisms. Further studies relying on multi-parametric, morphometric and functional assessments of small nerve fibres are needed to better clarify the relationship between symptoms, and histopathological and functional abnormalities. 

### 4.2. Autonomic Small Fibre Impairment

Patients with fibromyalgia commonly complain of autonomic disturbances, such as orthostatic intolerance, flushing and thermal intolerance, and digestive and sudomotor abnormalities [86]. Although different hypotheses have been formulated to explain autonomic disturbances in fibromyalgia [87,88], such as abnormal hypothalamic–pituitary axis functioning, the evidence of small fibre pathology in patients with fibromyalgia has strengthened the hypothesis that autonomic small fibre impairment could contribute to autonomic symptoms in these patients [89]. 

Our systematic review also included studies that evaluated dysautonomia using different objective functional testing methods. In line with a previous systematic review [90], we found that most studies based on HRV analysis demonstrated attenuated HRV and an increased LF/HF ratio, which is associated with high sympathetic tone and low parasympathetic activity [91]. HRV is one of the most widely used functional measures to assess autonomic nervous system impairment in patients with small fibre neuropathy [92,93], and its blunt impairment in fibromyalgia could be partially subtended by small fibre damage. However, HRV parameters do not solely depend on peripheral autonomic nerve fibre activity and are largely influenced by central nervous system modulation [94], thus implying that HRV abnormalities cannot be considered as unequivocal signs of autonomic small fibre impairment. 

Studies based on sympathetic skin response, tilt testing, and skin conductance have demonstrated more controversial results, with only a slight majority of the studies for each technique supporting autonomic small fibre impairment in patients with fibromyalgia. Sympathetic skin response and skin conductance are simple, quick, and non-invasive tests that are frequently used to diagnose functional impairment of sudomotor sympathetic fibres in peripheral neuropathies [95,96]. However, they suffer from the same limitations as HRV analysis, since their responses are not entirely mediated by peripheral autonomic nerve fibres and are largely influenced by emotional, attentional, and environmental factors [95,97].

Our findings show that highly prevalent autonomic dysfunction can be detected in fibromyalgia patients using different autonomic function tests, investigating different autonomic domains, including cardiovascular and sudomotor domains. However, the role of small fibre impairment in autonomic symptom pathogenesis in fibromyalgia is still unclear. Further studies are needed, using tests that selectively explore peripheral autonomic nerve fibres, such as QSART and skin biopsy with the quantification of autonomic innervation, and using methods that are able to distinguish between pre- and post-ganglionic autonomic damage, such as myocardial scintigraphy or eye drop testing with pilocarpine/phenylephrine. 

Another open question is how small fibre pathology influences symptoms, such as fatigue and exercise intolerance. As our results show, laser Doppler flowmetry studies have documented a blood flux reduction primarily over the tender points in patients with fibromyalgia, thus suggesting that excessive fatigue could be caused by peripheral tissue ischemia and hyperactivation of deep tissue nociceptors by anaerobic metabolites and inflammatory cytokines [83,84]. It has recently been suggested that blood flow dysregulation, as a result of excessive innervation to arteriole-venule shunts, could contribute to widespread deep pain and fatigue in fibromyalgia. The excessive innervation involves a greater proportion of vasodilatory sensory fibres than vasoconstrictive sympathetic fibres [26].

## 5. Conclusions

Our systematic review shows that approximately 50% of patients with fibromyalgia show signs of somatic and autonomic small fibre impairment, further supporting the evidence of frequent small fibre pathology in patients with fibromyalgia. 

However, while skin biopsy studies from different research teams were consistent in showing that a considerable number of patients had small fibre pathology, studies that used functional investigations to assess somatic and autonomic small fibre function provided conflicting results, thus challenging small fibre pathology as the leading mechanism underlying fibromyalgia’s painful and autonomic symptoms. 

## Figures and Tables

**Figure 1 diagnostics-12-01135-f001:**
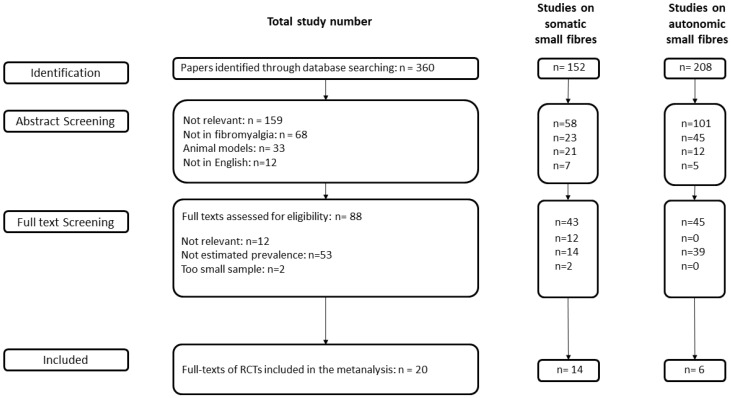
Flow chart of the search process of papers included in the meta-analysis.

**Figure 2 diagnostics-12-01135-f002:**
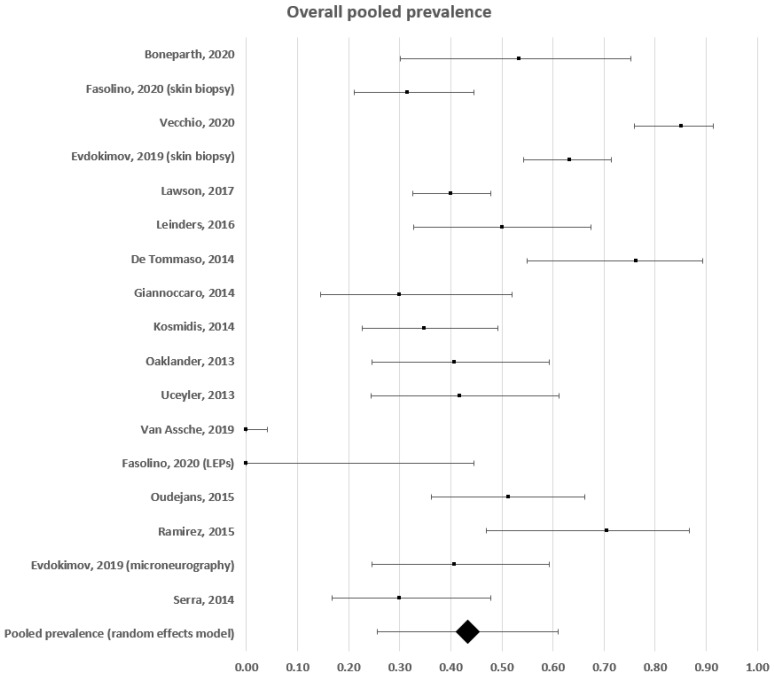
Forest plot showing overall pooled prevalence estimates of small fibre impairment in fibromyalgia.

**Figure 3 diagnostics-12-01135-f003:**
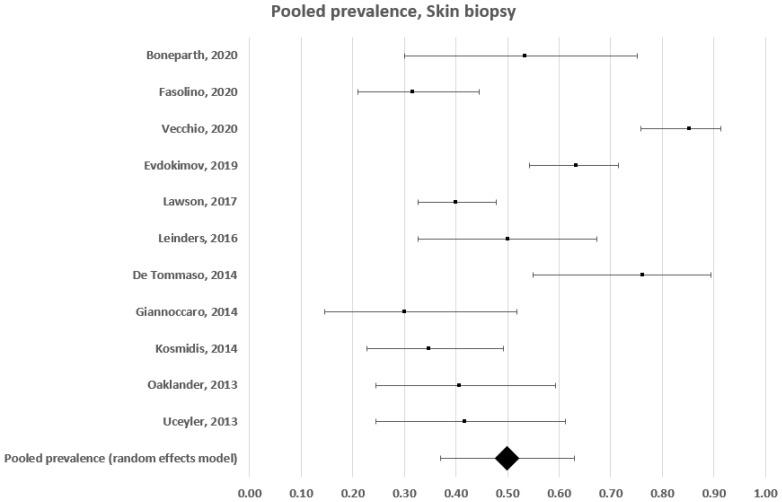
Forest plot showing pooled prevalence estimates of small fibre impairment in studies using skin biopsy.

**Figure 4 diagnostics-12-01135-f004:**
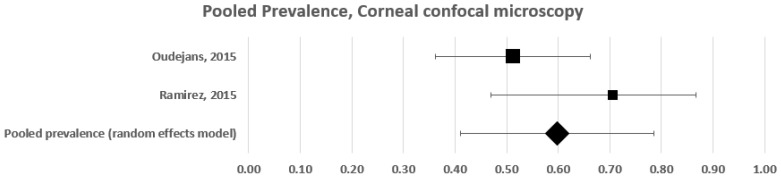
Forest plot showing pooled prevalence estimates of small fibre impairment in studies using corneal confocal 165 microscopy.

**Figure 5 diagnostics-12-01135-f005:**
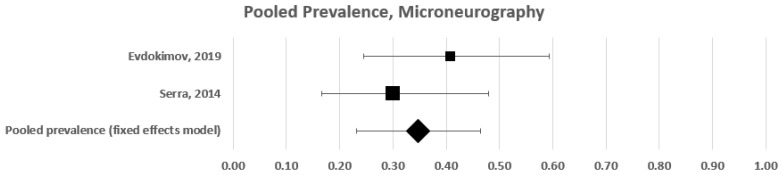
Forest plot showing pooled prevalence estimates of small fibre impairment in studies using microneurography.

**Figure 6 diagnostics-12-01135-f006:**
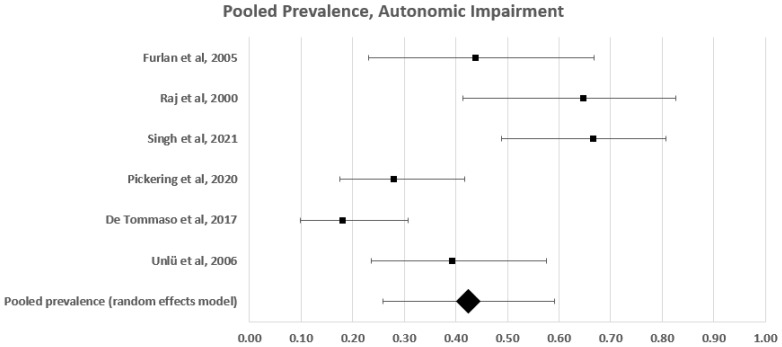
Forest plot showing pooled prevalence estimates of autonomic disfunction.

**Table 1 diagnostics-12-01135-t001:** Studies dealing with somatic small fibre impairment included in the meta-analysis.

Reference	Sample Size	Fibromyalgia	Control Group	Prevalence Number	Prevalence Estimate (%)
**Skin biopsy**					
Boneparth 2021 [15]	38	15	23	8	53
Fasolino 2020 [16]	57	57	-	18	32
Vecchio 2020 [17]	81	81	-	69	85
Evdokimov 2019 [18]	128	117	11	74	63
Lawson 2018 [19]	155	155	-	62	40
Leinders 2016 [20]	116	28	88	14	50
De Tommaso 2014 [21]	81	21	60	16	76
Giannoccaro 2014 [22]	52	20	32	6	30
Kosmidis 2014 [23]	80	46	34	16	35
Oaklander 2013 [24]	57	27	30	11	41
Uceyler 2013 [25]	155	24	131	10	42
**Corneal confocal microscopy**					
Oudejans 2016 [32]	*	39	*	20	51
Ramirez 2015 [8]	34	17	17	12	71
**Microneurography**					
Evdokimov 2019 [18]	41	27	14	11	41
Serra 2014 [9]	56	30	26	9	30

* Information not available.

**Table 2 diagnostics-12-01135-t002:** Studies dealing with autonomic small fibre impairment included in the meta-analysis.

	Sample Size	Fibromyalgia	Control Group	Prevalence Number	Prevalence Estimate (%)
**Tilt test**					
Furlan 2005 [64]	32	16	16	7	44
Raj 2000 [67]	31	17	14	11	65
**Heart rate variability**					
Singh 2021 [42]	60	30	30	12	40
Lee 2016 [49]	60	35	25	29	83
**Skin conductance**					
Pickering 2020 [73]	100	50	50	14	28
**Sympathetic skin response**					
De Tommaso 2017 [38]	80	50	30	9	18
Ünlü 2006 [82]	46	28	18	11	39

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
