# Peer review of "A Systematic Review and Meta-Analysis of the Prevalence of Small Fibre Impairment in Patients with Fibromyalgia"

_diagnostics, 2022, doi:10.3390/diagnostics12051135_

Round 1
Reviewer 1 Report
This manuscript presents a systematic review and meta-analysis related with the prevalence of small fibre impairment in patients with fibromyalgia. This review is generally well written, but it is unclear what this review adds to what is already known and have been published earlier. No clear research question seems to be formulated, the conclusions are unclear and other major concerns with this manuscript.
My specific comments are stated below. Overall, several important issues need to be addressed and some are of methodological character which requires a considerable revision of the paper.
Abstract: The abstract should be a total of about 200 words maximum. The abstract should be a single paragraph and should follow the style of structured abstracts, but without headings: 1) Background: Place the question addressed in a broad context and highlight the purpose of the study; 2) Methods: Describe briefly the main methods or treatments applied. Include any relevant preregistration numbers, and species and strains of any animals used. 3) Results: Summarize the article's main findings; and 4) Conclusion: Indicate the main conclusions or interpretations. The abstract should be an objective representation of the article: it must not contain results which are not presented and substantiated in the main text and should not exaggerate the main conclusions.
Introduction
1. The introduction section did not provide a clear rationale for carrying out the study (for example, why is your research question important? What gap in the literature is the study addressing?
I suggest in this section in the first paragraph should have a sentence or two added that better outlines why this study is important related with quality of life related to foot health status in women with fibromyalgia, to see the research of Palomo et al https://pubmed.ncbi.nlm.nih.gov/31110536/
In the last paragraph, the significance of the proposed word should be included highlighting why your work is important. what is the scientific contribution of this paper? it is not clear how this paper can make a significant contribution to the state of the art.
In addition, author´s hypotheses should be included .
2. Methods: Was the study registrered at PROSPERO? Please to include the ID - number.
3. Methods - literature search and selection: Please outline the exact search string or provide an appendix with the search strategy with specific search outcomes for each search and combinations.
4. Methods - literature search and selection: Did you restrict study selection on any language?
5. Results.The authors have not performed a systematic review, according to international standards, so they do not provide specific numerical data. Please add the calculate related with risk of bias was evaluated of this investigation.
6. Within your discussion, compare outline your results, discuss their novelty and their application to practice.
7. Conclusion. These conclusions need to be softened, modified a in order to reflect only the study findings.
Reviewer 2 Report
Galosi, Truini, and Di Stefano systematically reviewed and meta-analyzed the prevalence of small fibre impairment in patients with fibromyalgia. The presentation is clear and logically built up; both Results and Discussion sections will benefit further structuring. Please find comments and proposed revisions below for further improvement of this nice systematic review and meta-analysis.
Line 11: Please add “We searched PubMed, EMBASE, and Cochrane Library databases systematically, and we found 350 articles…”
L.27: more often than males
L.48-56: details on the three searches should be documented as Supplemental Material to increase transparency of the work done.
- 64 and Fig. 1 (flowchart) belong both to the very beginning of the Results section.
L.70, l.84: It is usual practice to exclude studies with less than 10 patients included (l.56-57), but there is no reason why studies estimating the prevalence as 0 should be excluded. Also 0 out of n is an estimable proportion (0) with a respective 95% confidence interval (95% CI). I can easily see that your approach for 95% CI estimation will fail as you employed Wald-type CIs. Further comments to this end follow below.
L.71-73: (Approximate) Wald-type 95% CI as employed here can and are often used for continuous outcomes. For binary outcomes, like the prevalence here (which is a proportion x out of y), Wald-type 95% CI are known for insufficient coverage probabilities when n is small and/or p is close to the lower (0) or upper (1) boundary for proportions. A Wald-type 95% CI will, for instance, for an estimated prevalence of 0 comprise negative values – which simply do not make sense. One solution is to compute 95% CI as ‘exact’ Clopper-Pearson 95% CIs; alternatively, Wilson score-type 95% CI can be used which are less conservative than exact ones and were proposed earlier to be used instead of exact 95% CI. See, for instance, Chapter 6 “Proportions and their differences” in Altman DG, Machin D, Bryant TN, Gardner MJ, Statistics with Confidence, 2nd ed. Bristol, UK: BMJ Books, 2000.
L.79: Please add significance level (5%, two-sided) and the software package that you used for your calculations.
L.119, 126, 169, 190-192, 253, 268: please keep increasing order for cited references (e.g. [8, 29], not [29, 8])
Table 1: sample size for Evdokimov 2019 should probably read 128. Prevalence estimates for Kosmidis 2014, Evdokimov 2019, and Serra 2014 are 35, 41, and 30, respectively, given the numbers for fibromyalgia and prevalence number. Please correct and recheck your calculations, thanks.
L.143, 151, 161, 171, 228: replace “Forest plot analysis showed that the random effects overall frequency was” by “Forest plot analysis showed that the estimated prevalence, based on a random effects model, was”
L.145, 154, 163, 173: replace “P < 0.00001).” by “P < 0.0001).”
Fig. 2, Fig. 3, Fig. 4, and Fig. 6: replace “Pooled Prevalence (Radom Model)” by “Pooled prevalence (random effects model)”. X-axis should read 0, 0.10, 0.20, 0.30, 0.40, 0.50, 0.60, 0.70, 0.80, 0.90, 1 (use colons, not commas, restrict range to 0 to 1) to make Forest plots for different analyses easier to compare with each other visually.
Table 2: Round all prevalence estimates to integers (for the sake of consistency). Please correct Unlu¨ et al., 2006
Fig. 5: X-axis should read 0, 0.10, 0.20, 0.30, 0.40, 0.50, 0.60, 0.70, 0.80, 0.90, 1. Please correct Unlu¨ et al., 2006
L.148, 157, 167: renumber sections on subgroup analyses to 3.1.1, 3.1.2, and 3.1.3, respectively.
- 173: replace 0.000 by 0
L.177: 3.5 becomes 3.2
L.218: integrate this section into the previous one.
Fig. 5: replace “Pooled Prevalence (Fixed Model)” by “Pooled prevalence (fixed effects model)”
Discussion section: Please structure the Discussion, for instance, according to Docherty M, Smith R. The case for structuring the discussion of scientific papers. BMJ. 1999 May 8;318(7193):1224-5. doi: 10.1136/bmj.318.7193.1224. They proposed
- Statement of principal findings
- Strengths and weaknesses of the study
- Strengths and weaknesses in relation to other studies, discussing particularly any differences in results
- Meaning of the study: possible mechanisms and implications for clinicians or policymakers
- Unanswered questions and future research
Your very first paragraph (l.235-238) could function as “Statement of principal findings”, and the current section 4.1 and 4.2 can become part of “Strengths and weaknesses in relation to other studies…”.
Round 2
Reviewer 1 Report
The paper has much improved, and although I have reservations about the interpretation of the data, and the strength of evidence for the clinical message, I think the article presents the data well enough for readers to judge themselves. I would recommend publication.